# BIM-Based Digital Twin and XR Devices to Improve Maintenance Procedures in Smart Buildings: A Literature Review

**Corentin Coupry** [1,2,*], **Sylvain Noblecourt** [2], **Paul Richard** [1], **David Baudry** [3] **and David Bigaud** [1]

1   LARIS, SFR MATHSTIC, Univ. Angers, F-49000 Angers, France; paul.richard@univ-angers.fr (P.R.); david.bigaud@univ-angers.fr (D.B.)
2   LINEACT, CESI, 44 Avenue Frédéric Auguste Bartholdi, 72000 Le Mans, France; snoblecourt@cesi.fr
3   LINEACT, CESI, 80 rue Edmund Halley, 76808 Saint-Étienne-du-Rouvray, France; dbaudry@cesi.fr
*   Correspondence: corentin.coupry@univ-angers.fr; Tel.: +33-607943470

**Abstract:** In recent years, the use of digital twins (DT) to improve maintenance procedures has increased in various industrial sectors (e.g., manufacturing, energy industry, aerospace) but is more limited in the construction industry. However, the operation and maintenance (O&M) phase of a building's life cycle is the most expensive. Smart buildings already use BIM (Building Information Modeling) for facility management, but they lack the predictive capabilities of DT. On the other hand, the use of extended reality (XR) technologies to improve maintenance operations has been a major topic of academic research in recent years, both through data display and remote collaboration. In this context, this paper focuses on reviewing projects using a combination of these technologies to improve maintenance operations in smart buildings. This review uses a combination of at least three of the terms "Digital Twin", "Maintenance", "BIM" and "Extended Reality". Results show how a BIM can be used to create a DT and how this DT use combined with XR technologies can improve maintenance operations in a smart building. This paper also highlights the challenges for the correct implementation of a BIM-based DT combined with XR devices. An example of use is also proposed using a diagram of the possible interactions between the user, the DT and the application framework during maintenance operations.

**Keywords:** digital twin; BIM; augmented reality; mixed reality; maintenance; facility management; O&M





## 1. Introduction

Introduced in 2011 at Hannover Messe [1], Industry 4.0 (I4.0) define the new Industrial Revolution. Following the previous ones, which lasted almost 200 years, it promotes the digitalisation and automation of industrial processes to improve their quality and reliability. One of its main goals is to improve the interoperability and readability of machines and processes, as well as to permit decentralising support and decision-making [2]. In the construction industry, which is the least digitalized (only 1% of its revenue is invested in research [3]), this revolution is often known as Construction 4.0 [4]. To achieve these goals, it is needed to implement new methodologies and technologies (e.g., IoT (Internet of Things) platforms, mobile devices, smart sensors, cloud computing). Among these, the most explored in recent years are Digital Twins (DT) and extended reality technologies (Augmented Reality (AR), Mixed Reality (MR) and Virtual Reality (VR)), defined under the generic name of XR [5,6].

One of the most expensive industrial processes today is maintenance. In the construction industry, the Operation and Maintenance (O&M) phase represents the largest part of a building's lifecycle. Mourtzis et al. [7] observes that 30% of the total management costs are associated with maintenance procedures. Each industrial revolution has brought more complexity to existing processes, requiring new procedures to be devised

(e.g., maintenance and safety [8]). For I4.0, the new maintenance paradigm is known as *Maintenance 4.0* or *Smart Maintenance*. According to Jasiulewicz-Kaczmarek et al. [9], this new paradigm uses the knowledge from I4.0 tools to implement proactive strategies, such as failure prediction, fatigue estimation or energy consumption optimisation. These new strategies can be assimilated to *predictive maintenance* or even *prescriptive maintenance* when it also suggests how maintenance can be performed. According to McKinsey [10], these strategies can provide up to USD 630 billion in potential savings in maintenance operations in 2025.

Many research projects have been conducted using I4.0 tools to improve maintenance operations, such as BIM (Building Information Modeling), digital twin and XR devices. The BIM concept is derived from 3D CAD (Computer-Aided Design) models and may be seen as the evolution of BDS (Building Description System) proposed by Eastman in 1974 [11]. The BIM model is often considered as a graphical database of a building, containing its geometric and semantic data, which allows access to data throughout its entire lifecycle directly from the equipment [12–14]. This data access, and the fact that they can contain all the geometric and semantic data of the building, has led many researchers to try to use BIM as a basis to develop the DT of a building, for either existing buildings or those in the design stage [15,16].

The concept of Digital Twin first appeared in 2002 in a presentation by Grieves on the concept of PLM (Product Lifecycle Management) as an evolution of it, and it was then redefined in 2012 by Grieves and Vickers [17] as a digital representation of a physical asset and the automatic connections that bind them together. The authors highlight that the data set it refers to must describe an asset in a comprehensive way, from its geometric characteristics to its operational records. The definition of Glaessgen and Stargel [18] is presently the most commonly shared definition: "*An integrated multiphysics, multiscale, probabilistic simulation of an as-built vehicle or system that uses the best available physical models, sensor updates, fleet history, etc., to mirror the life of its corresponding flying twin*". They emphasise the data management of a DT and the importance of a real-time connection to ensure twins' reliability. One of the main purposes of the DT is that it can use the data collected in real-time and historical data to perform simulations on the future state of an asset and to predict the consequences of maintenance operations for optimisation purposes. Aeronautical and aerospace industries were the first industries interested in the use of DTs to perform remote monitoring on their assets [18,19].

It has also been observed that there is no clear definition for a DT [20]. Kritzinger et al. have observed that some so-called DT projects do not entirely respond to the main definition of a DT and proposed three main sub-classes of so-called DT [21]. The first one is the *Digital Model* or DM, very similar to a BIM when used in the construction industry, corresponding to a digital representation without automatic data transfer. They also defined the *Digital Shadow* or DS, where the data are automatically sent in real-time from the physical to the digital asset only. Their *Digital Twin* is then defined as a DM with automatic exchange both from the physical to the digital asset and vice versa (Figure 1). Some authors, such as Aheleroff et al. [22], have defined a fourth sub-class of DT: the *Digital Twin Predictive*, which is defined as a DT using predictive algorithms for further analysis. We decided to include this fourth sub-class in the *Digital Twin* sub-class, in line with the main characteristics of DTs given by Grieves [17] and Glaessgen and Stargel [18]. These subtypes can be used to observe the advancement in the development and challenges that need to be overcome to create a *true* Digital Twin. These different observations allow us to establish our own definition of what a true DT should be: *A Digital Twin is a multi-scale representation of a whole consisting of a potential or existing system (physical product, user and activity) in the real environment, its virtual reflection in the digital space and the processes of automated exchange of data and information in real-time and using simulation algorithms and historical data or that collected from smart sensors to predict the system's future state or its response to a given situation. A Digital Twin may also include the Digital Twins of its subsystems.*

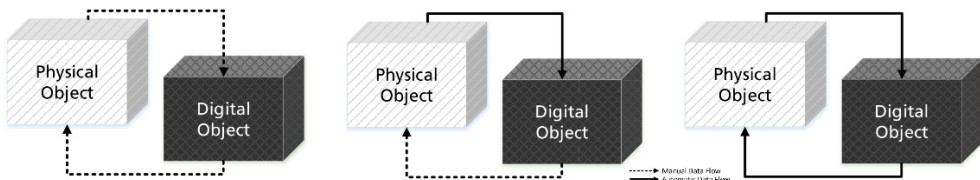

**Figure 1.** Digital Model (**left**), Digital Shadow (**middle**), Digital Twin (**right**) [21].

One of the main revolutions brought by the I4.0 is the introduction of the XR technologies. The term XR defines a spectrum containing AR, MR and VR paradigms, from the least to the most immersive one. The term *augmented reality* is first introduced in 1992 by Caudell and Mizell [23] to define virtual objects as overlapping with their users' real environment thanks to a specific device. The VR devices are mainly head-mounted display systems that allow the user to be immersed in an interactive virtual environment. According to Gartner, the use of these technologies has increased in the last year, thanks to the recent performance improvements and the miniaturisation of mobile devices. These improvements have allowed the implementation of MR devices, which were first defined in 1994 [24] and can be seen as improved AR devices with which the user can interact both with the digital and physical elements using intuitive interaction methods (e.g., information appeared according to user's position in space [25], take control of physical equipment through the use of the device [26]).

Numerous research studies have been conducted to show the benefits of using BIM or XR devices to improve O&M in the construction industry, but also how their combined use can help to resolve some challenges. Furthermore, as observed in a literature review performed by Errandonea et al., DT can bring many improvements to maintenance operations in industrial processes [19]. However, their review does not address the uses of BIM and XR technologies with a DT or the specific benefits to the construction industry. Research about the use of DT is still in its early stages of development, as no practical applications were found before 2017 [19,27]. We have also observed that there are few if any papers using these technologies in combination.

In this paper, we will study how the BIM and XR devices can further improve the development and the use of a DT to improve maintenance operations in the construction industry, from planning to work instructions. The paper is organised as follows: Section 2 describes the methodology used to find the papers that we analyse in the following sections. Then, we will focus on the benefits and challenges of the use of a BIM model to create a DT in Section 3 and on the various improvements such a DT can bring to maintenance operations in Section 4. Then, we see how XR technologies can help to visualise and interact with the DT in Section 5 and the different improvements the use of these two technologies combined can bring to maintenance operations in Section 6. In these three sections, we explore both the DT developed with and without a BIM model as a basis. Finally, we categorise and analyse the different articles that we found and we highlight some perspectives and challenges in Section 7.

## 2. Methodology

In order to evaluate the use of BIM-based DT and XR technologies to improve maintenance operations, a systematic literature review methodology was used. The main aim is to observe how these technologies are used in the building industry for maintenance purposes and to identify the gaps in the literature. Then, this research can be used to provide future fields of research.

Multiple databases were selected to perform this research:

- ScienceDirect (https://www.sciencedirect.com/) (accessed on 5 February 2021)
- Google Scholar (https://scholar.google.fr/) (accessed on 5 February 2021)
- WebOfKnowledge (https://www.webofknowledge.com/) (accessed on 5 February 2021)
- Scopus (https://www.scopus.com/) (accessed on 5 February 2021)

Moreover, a manual search of Grey Documentation was performed to avoid relevant papers available on the Internet and published by non-academic institutions. This research was performed using the software Harzing's Publish or Perish (https://harzing.com/resources/publish-or-perish (accessed on 5 February 2021)), which helps to perform a cross-database research. The reference manager software utilised was Zotero (https://www.zotero.org/ (accessed on 5 February 2021)) due to its large community and the automatic citation add-in for Microsoft Word.

We decided to focus the research on the combined use of the terms "*Maintenance*", "*BIM*", "*Extended Reality OR XR*" and "*Digital Twin*". This research was conducted over a time window from 2005 to February 5, 2021, to observe the most recent developments and implementation of these technologies. A Venn diagram of the terms combinations has been created (see Figure 2). The size of the different coloured areas does not correspond to the number of items found for each combination but helps to identify the different combinations used. The different numbers are used to show the various combinations used: 1 for "*Maintenance*", 2 for "*BIM*", 3 for "*Extended Reality OR XR*" and 4 for "*Digital Twin*" However, to avoid missing any relevant paper, the term "*Extended Reality OR XR*" was broken down in its various subtypes: "*Augmented Reality OR AR*"; "*Mixed Reality OR MR*" and "*Virtual Reality OR VR*".

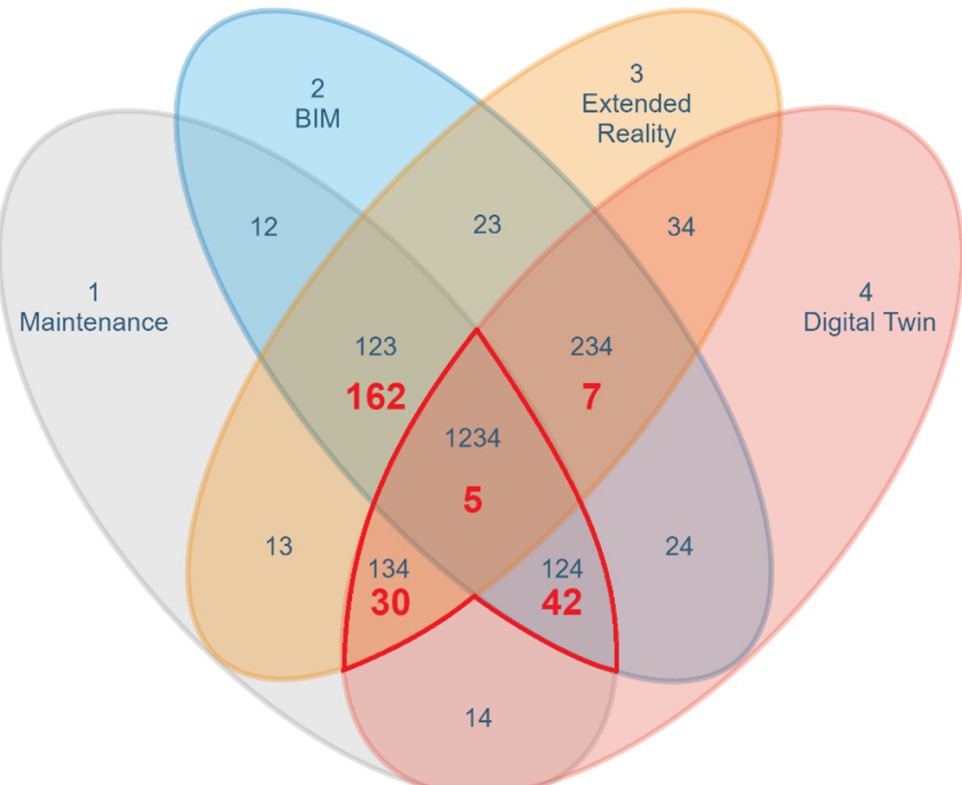

**Figure 2.** Venn diagram of the terms combinations used for our research. The red outline delimits the combinations selected for our research. The red numbers represent the number of papers found for the combination of three and four terms.

In the selected period, many papers were published for some of these combinations (e.g., More than 1200 only for the combination "*XR AND Maintenance*"). On the other hand, some of these combinations very few results (e.g., 5 results on the overall combination "*BIM AND Digital Twin AND XR AND Maintenance*"). The research was therefore extended to the papers using a DT based on a BIM for maintenance purposes and to the DT using XR devices for maintenance purposes. The combinations used for this research are outlined in red in Figure 2:

- *"BIM" AND "Digital Twin" AND Maintenance* (combination *"124"* in Figure 2). This combination represents 42 papers.
- *"Digital Twin" AND ("Extended Reality" OR XR) AND Maintenance* (combination *"134"* in Figure 2). This combination represents 30 papers.
- *"BIM" AND "Digital Twin" AND ("Extended Reality" OR XR) AND Maintenance* (combination *"1234"* in Figure 2). This combination represents 5 papers.

To avoid missing any relevant paper, the term "*Extended Reality OR XR*" was again broken down into its various subtypes ("*Augmented Reality OR AR*"; "*Mixed Reality OR MR*" and "*Virtual Reality OR VR*"). Once the bibliographical research had been performed, inclusion and exclusion criteria were applied to the different strings to narrow down the results to the most pertinent papers (see Table 1).

**Table 1.** Inclusion (IC) and exclusion (EC) criteria.

| Inclusion criteria (IC) | Main subject is the use of DT (concept or use case) | Exclusion criteria (EC) | Older than 2005 |
|---|---|---|---|
| | | | Not in informatics or engineering field |
| | Use of BIM and/or XR technologies | | Not related or applicable to building maintenance |

We decided to keep the papers in which the use of DT is the focus and in which it is used in combination with at least a BIM or XR technology. Although the search was for combinations of these terms, some papers that did not use these technologies came up and were excluded by this criterion. We also decided to include papers that were not specific to the construction industry but whose methodology or results could be transposed to it. The criteria were applied separately for each searching string to narrow down their respective numbers. First, they have been applied during the initial result of 242 papers before excluding the duplicates of the different databases. Then, the criteria were applied to the title, abstract and introduction of each remaining paper before merging the different strings results. Finally, the new duplicates, due to the different research strings, were excluded. This methodology can be observed in Figure 3. The result of this methodology is a pool of 68 relevant papers, including conceptual projects, use cases and reviews.

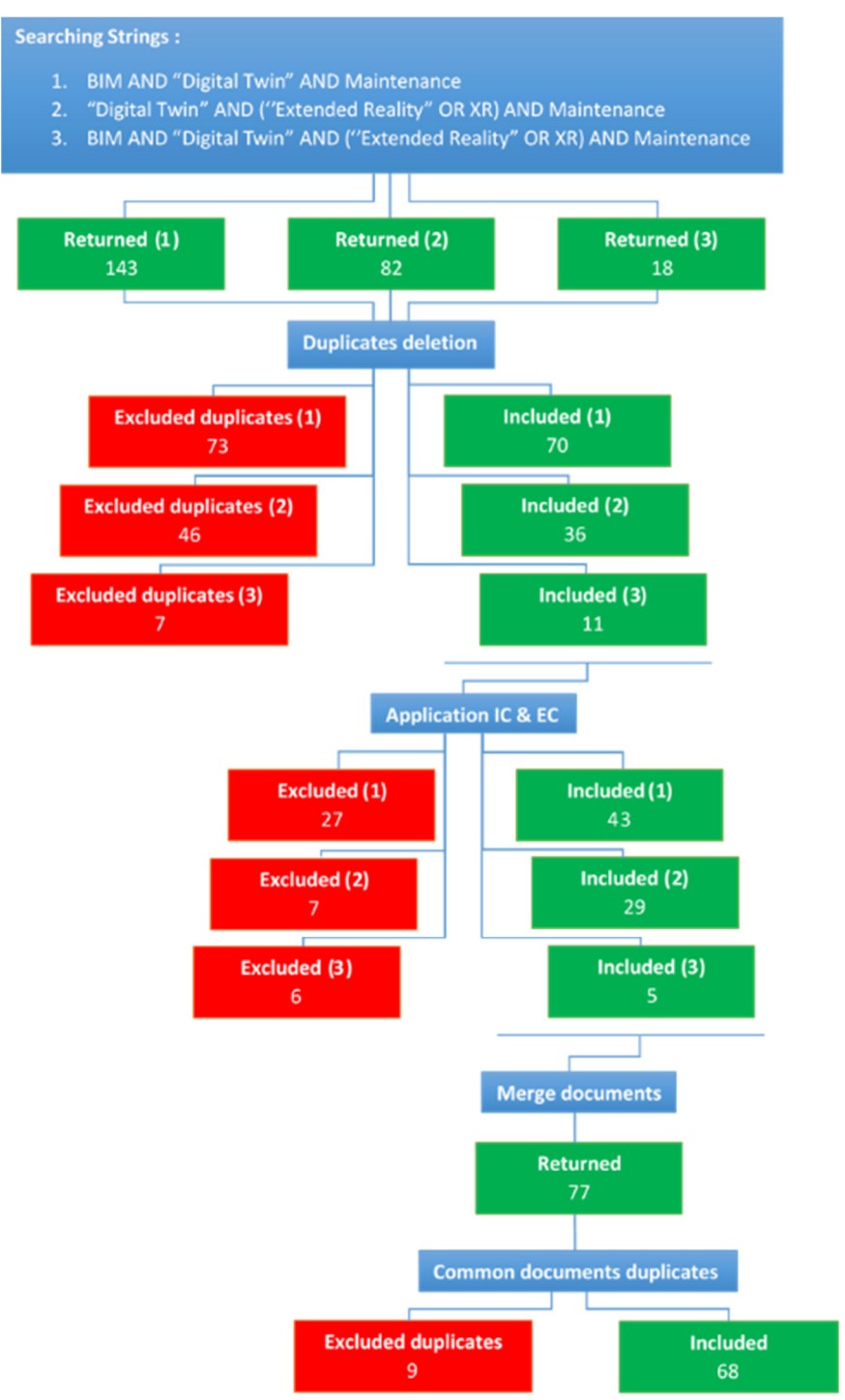

**Figure 3.** Selection of studies for primary studies for systematic review starting from 243 papers.

## 3. Benefits of a BIM-Based DT

Based on the 68 papers found in this research, we identified 17 papers that highlight the benefits of using a BIM to create a DT. In the literature, we observed that the concept of using a BIM to develop a DT has been widely explored (see Table 2) and has begun to be used in use cases (more than 62% of the identified papers are conceptual papers). Because BIM is mainly used in the early phases of the project, it can contain precious information for building exploitation.

**Table 2.** BIM as a basis for creating a DT for the construction industry.

| Research Topics | Benefits | Reference |
|---|---|---|
| Usage of a BIM-based DT | Early optimisation of the building<br>Lifecycle management of the building<br>Condition assessment<br>Optimisation of existing operations | [28–30]<br>[16,31–36]<br>[37–39]<br>[39–42] |
| Creation of a DT using a BIM | DT as an evolved BIM with real-time data<br>BIM provide geometrical data/3D model<br>BIM contains static data | [29,32,35,37–40,42]<br>[16,29–31,33,35,37–42]<br>[28,35,40,42] |
| Benefits of a BIM-based DT for lifecycle management | BIM mainly used in early stages of the lifecycle<br>Simulations/Predictions of building lifecycle<br>Improve decision-making<br>Improve operations | [16,28,40]<br>[28,29,31,41]<br>[28,29,31–35,37–42]<br>[32,33,36,40] |
| Improvements for data management | Centralised source/Data sharing<br>Digital continuity/Interoperability<br>Data linked to models | [16,29,32,33,38,40,41]<br>[16,28,29,32,34–37,40,41]<br>[32–34,37,41,43] |

### 3.1. Help to Create a DT

Originally, BIM is mainly used in the early phases of a project to ensure it is consistent with architectural plans. In the design phase, the BIM can be used to provide a 3D representation of building components [16,29–33,35,37–42]. For example, in 2020, Sofia et al. created a BIM model of a cable bridge to obtain its geometric characteristics in order to include it in an urban environment and observe its performance [44]. The BIM can also be used to provide ontological information [35], which can then be used to perform simulations to provide early optimisation of the building construction and exploitation (such as energy consumption [29,35] or carbon emission [32,37,38]). However, originally, BIM is mainly constructed to provide static data, such as documentation and ontological and geometrical information [28,35,40,42]. Some authors have proposed to integrate real-time data using smart sensors (e.g., accelerometers, inclinometers and temperature sensors) to improve the reliability of the BIM, helping it to evolve towards a better representation of the environment [29,32,35,37–40,42]. For example, Angjeliu et al. installed smart sensors on critical points of the Milan Cathedral to evaluate the evolution of its health [39]. By providing analytical algorithms to manage the collected data and to link it to its corresponding elements, we can observe that this evolved BIM comes close to the main DT definition.

### 3.2. Usage in Lifecycle Management

While the main purpose of BIM is to help in the creation phase of the building, combined with prediction algorithms, the static data it contained can already be used in early phases to perform simulations to further optimise the exploitation of the building or equipment placement [16,28,29,31,40,41]. Bolshakov et al. have observed, for example, that if the actors in the O&M phase of a system are involved in its design phase, it is then possible to set up test operations early enough to avoid costly modifications in the future [28]. Such data can also be used in the construction phase to detect any misalignment in the building elements by using it as a comparison basis with the constructed element, in order to avoid any further necessary reparation [45]. A BIM-based DT can also be used to improve existing operations in the building lifecycle, such as optimising equipment usage or global maintenance costs [32,33,36,40]. Kaewunruen et al. show the example of using an enhanced BIM to improve the management and the maintenance cost of a railway turnout system according to its condition and track usage [32].

### 3.3. Improvement for Managing Data

Research studies have been done to highlight that, if used from the start of the project, BIM can contain static and semantic data of the entire project, such as providing digital continuity throughout its lifecycle [16,28,40]. To ensure this continuity and the interoperability between the different phases of the project, some authors have proposed

methodologies and points of concern to prevent any loss of data between the different phases [16,28,29,32,34–37,40,41]. Prušková et al. propose such a methodology to explain how the data are transferred throughout the building lifecycle [36]. It has been shown that the centralised property offered by a singular BIM-based DT can improve data access by the different stakeholders of the project [16,29,32,33,38,40,41]. Oliveira highlights how such centralisation can help to prevent missing or conflicting information during the management of an airport [41]. Through this centralisation, they can provide their own knowledge and expertise for further analysis. The 3D representation of physical equipment provided by the BIM can be used to improve the clarity and accessibility of the data once it has been linked to their corresponding equipment [32–34,37,41,43]. Xie et al. show how this link can help access data during inspection procedures, for example [43]. Once the data are linked, the user can obtain all the knowledge gathered about the specific equipment they are focusing on. These various improvements in data accessibility and expertise provide important support when decisions have to be made (e.g., choosing the maintenance procedure to be performed, identifying the faulty equipment) [28,29,31–35,37,38,40–42].

### *3.4. Existing Challenges*

The use of a BIM-based DT brings new challenges to overcome in order to ensure its reliability, as we can see in Table 3. One of the main issues concerns the lack of up-to-date data [28,33], usually caused by incorrectly anticipating the need of later phases and by a lack of communication between the different actors in these phases. The stakeholders often use specific BIM models for their current phase [16], making it necessary to improve the formalisation and standardisation of data to provide interoperability between the phases but also between the various systems inside the building [16,28,31,33,35,36]. As an example, Peng et al. have observed that the specificity of specific equipment, such as medical systems included in the BIM, complicates the automatic data recuperation and analysis, but also the data exchange between the different phases of the project [35]. Moretti et al. proposed a methodology to ease the implementation of real-time data by creating objects in the IFC standard, originally defined as a BIM standard [34].

**Table 3.** Challenges of BIM-based DT creation and exploitation.

| Challenge | Reference |
| --- | --- |
| Lack of standards | [16,28,31,33,35,36] |
| Lack of up-to-date data | [28,33] |
| Privacy issues | [29,31,35] |
| Lack of organisational strategy | [29,35,36,41] |

The improved sharing of data also provides an issue concerning the security and the privacy of data [29,31,35], concerning who can see and use the collected data and who is proprietary of these data. It brings about the need to provide new organisational strategies concerning the roles of each stakeholder, but also on the implementation of the BIM-based DT into the existing strategies of the building exploitation [29,35,36,41].

### 4. Maintenance Improvements Bring by DT into O&M Phase

In the literature, we only found a single review on the usage of DT for maintenance in various industrial environments [19], but it may be relevant to observe if the benefits DT brings to the industry can also be applied to the exploitation of a building, whether in conceptual or concrete form (39% of the identified papers are use cases). Even if some of the DTs are not used to manage a building (25% of them are not BIM-based and focus on industrial structures or specific systems), we have observed that the improvements they bring to maintenance procedures can also be applied to building management (72% of the identified papers are using a BIM-based DT). We have identified 36 papers where these im-

provements are highlighted and studied. As we can see in Table 4, these improvements can concern any part of the maintenance procedures (e.g., monitoring, planning, inspection).

**Table 4.** Categorical review of the maintenance improvements with a BIM-based.

| Maintenance Step | Benefits | Reference |
|---|---|---|
| Monitoring | Better data access<br>Avoiding data silos<br>Optimise consumption<br>Collaboration | [16,22,26,34,35,39,43,46–53]<br>[22,33–35,51]<br>[29,37,38,40,54,55]<br>[26,34,44,51] |
| Inspection | Damage assessment<br>Deviation assessment | [39,42,47,56–59]<br>[42,45,60] |
| Planning | Early warning, prediction<br>Occupancy<br>Collaboration<br>Cost optimisation | [15,22,39,44,46,48,53,57,58,60–64]<br>[35,54]<br>[51,65]<br>[37,38,54,65] |

### 4.1. Improvements for Monitoring

One of the first improvements highlighted by the researchers concerns the global monitoring of the building and its equipment. Thanks to the centralised database provided by the DT, the different stakeholders obtain easier access to data related to the equipment they want to monitor [16,22,26,34,35,39,43,46–53]. For example, Peng et al. created a specific platform that allowed them to visualise in real time both the status of all the equipment in a hospital and the occupation of its different rooms [35]. This centralisation also helps to filter the large number of data that can be collected through the use of smart sensors [49,52] and to select meaningful data depending on the role of the user [47,51]. The specification of the data needed to effectively evaluate the system can be performed by the different actors of the project using the centralised DT. It allows for collaboration, allowing all the actors to contribute their own expertise in the data they need to effectively monitor the building [26,34,44,51]. Varé et al. have proposed a digital representation of a nuclear container that can be used by different teams (research, measurement and engineering) to exchange the information they need [51].

The centralisation of data also helps the stakeholders to monitor the building together and communicate their different analyses to the others by using the shared database and the link between the data and the equipment. This help in communication between the different actors throughout the project lifecycle provides the possibility to prevent isolation of data in their respective phases and thus a possible loss of important knowledge for further phases [22,33–35,51]. These benefits in monitoring can also help to optimise the global energy consumption and the exploitation costs of the building [29,37–40,54,55] during the inspection and maintenance planning phases. Kaewunruen et al. gave an example of these benefits during the reinforcement of a metro station taking into account the cost and environmental impact of the different materials that could be used [38].

### 4.2. Improvements for Inspection

Once the data collected by the smart sensors have been filtered and analysed, the DT's algorithms can be used to enrich the knowledge of the monitored system with automatic assessment abilities. Through the use of smart sensors combined with image capture devices, some authors have proposed methodologies to provide damage assessment abilities to the DT [39,42,47,56–59] to detect possible deterioration of the building. Shim et al. have proposed such a methodology to perform a visual assessment of deterioration on concrete bridges [59] and cable-supported bridges [58] using a coding system to inventory the identified deteriorations. Another problem that can have costly consequences and accelerate the deterioration of a building is the deviation of various elements in comparison with the initial conception of the building. To prevent further issues, some authors have proposed specific algorithms to detect possible deviation with their initial calibration [42,45,60]. Na-

hangi and Kim have presented an example of such an algorithm to quantify discrepancies and calculate realignments for construction components [45]. These algorithms are mainly useful for damage caused by unpredictable and uncontrollable events [17] and so are especially useful for reactive maintenance. However, some authors have also decided to use the predictive capabilities offered by the DT to improve proactive maintenance operations, which are the ones that are carried out to avoid the occurrence of a fault.

### 4.3. Improvements for Planning

Once inspection and damage assessment have been performed by the operators, it is necessary to plan the further maintenance procedures that must be done. We observed that many researchers have highlighted the improvements that specific and BIM-based DT can bring to such planning. Some authors have used prediction algorithms (e.g., machine learning, neural networks and linear regression) to visualise the later degradation and remaining life of equipment [15,22,39,44,46,48,53,57,58,60–64]. Tahmasebinia et al. use such algorithms to estimate the impact of long-term dead load creep and shrinkage on the Sydney Opera House [61]. With such prediction capabilities, the operators and managers can use the results to plan the maintenance early enough to prevent any equipment failure, helping them to move from *reactive maintenance* to *proactive maintenance*, such as *preventive maintenance* (Table 5). Some authors have also decided to use knowledge from previous incidents and maintenance reports to propose the optimised period in which the maintenance should be performed. This last improvement can be seen as *predictive maintenance*, where the maintenance planning is automatically performed and proposed to the operator. In Table 5, we can see which type of maintenance is the focus in the 68 papers we found. However, even with the term "*Maintenance*" included in all our searches, only 30 papers were found to provide this information.

**Table 5.** Maintenance types in the literature.

| Maintenance Type | | Reference |
|---|---|---|
| Proactive | Predictive | [15,22,37,40,43,44,46,47,49,52,55,63,64,66–69] |
| | Preventive | [34,39,57–60,62,64,65,70] |
| Reactive | | [25,55,71–73] |

Another advantage brought about by these algorithms is that they can be used to perform maintenance simulations to choose the most suitable one for the situation [18,63] or to compare the reparation cost with the equipment replacement cost [37,38,54,65] to propose the cheapest maintenance solution. Combined with predictive maintenance to perform automatic planning, these algorithms can be used to propose the best maintenance procedures to be performed to the operator, providing a *prescriptive* version of the maintenance [35,53]. The occupancy of the building is another type of data that can be used to optimise the proposed procedures [35,54], which can help with cleaning and maintenance procedures by preventing any interruption of tasks, or even to optimise energy consumption for unused rooms. The algorithms included in the centralised DT environment can be used by all authorised stakeholders to plan the maintenance procedures in a collaborative manner between the different professions [51,65].

### 4.4. Updating of the DT through Maintenance Operations

One of the emerging advantages of using a BIM-based DT during the inspection phase is that it allows the operator to observe whether there are any discrepancies between the DT and the actual condition of the equipment. Such differences can also be created by the operator when maintenance has been performed. Pileggi et al. [74] proposed a governance model called the Double Helix (see Figure 4) to represent the interactions between the DT and its physical counterpart through the addition of the role of a leading twin. This model allows for seamless switching between the twins as the situation requires, where one

leads while the other is updated (e.g., when the DT is leading, operations are performed on the physical system automatically or by the operator, and when the latter is leading, the DT is updated to reflect the new state of the system). This governance model can be used to improve digital continuity and to prevent further lack of up-to-date data following any repairs.

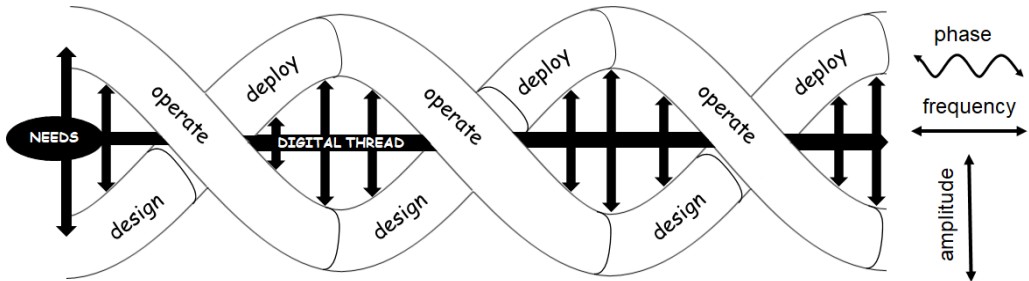

**Figure 4.** Double Helix model [74].

## 5. DT Improvements Brought about by eXtended Reality Technologies

The use of DT proves to be helpful to improve maintenance procedures. Nevertheless, it is necessary to provide users with an easy and intuitive interface to display the data included in the DT. The most studied technologies in the I4.0 to solve this problem are the eXtended Reality (XR) devices, such as Virtual Reality (VR), Augmented Reality (AR) and Mixed Reality [61,72,75,76]. In recent years, many researchers have proposed a conceptual or practical use of these technologies to improve the use of DT. Of the 68 papers collected for this research, we identified 23 that show us these improvements. However, only 39% of them are use cases. Interestingly, only 22% of these papers explore the use of a BIM-based DT. As we see in Table 6, we can observe that these devices can bring improvement to the visualisation and management of data but also provide new methods of interactions to the user.

**Table 6.** Categorical review of the improvements brought by XR.

| Functionalities and Technologies | Features | Reference |
|---|---|---|
| Management and data visualisation | Real-time and historical display<br>Situated display<br>Asset highlight<br>3D visualisation | [26,43,53,55,64,67,68,70,72,73,77,78]<br>[25,72]<br>[25,43,68,72,77]<br>[43,53,67–70,72,75,76,79] |
| Interaction with the model | Gesture and voice recognition<br>Standard inputs<br>Position tracking<br>Head/gaze tracking<br>MR control | [25,26,53,73,80]<br>[25,68–70,76–78]<br>[25,75,76]<br>[25,66,71–73]<br>[26,55,80] |
| Main devices used for visualisation | See-through HMD/ Smart-glasses (e.g., Hololens) (used for AR and MR)<br>Occluded HMD (e.g., HTC Vive) (used for VR)<br>HHD (e.g., Tablet; Smartphone) (used for AR) | [25,26,43,47,66,67,70–73,75,78,80]<br>[25,47,75,78,79]<br>[22,47,53,55,68,69,75,77,81] |
| Model update and collaboration | Update digital twin<br>Collaboration | [66,71,81]<br>[22,25,65,78] |

### 5.1. Management and Data Visualisation

One of the first characteristics of these devices is their ability to overlay information to the user in front of his or her surrounding environment. These visualised data can be called *augmentations*, because they provide deeper information to the operator concerning the performed operation. While benefits have already been observed in manufacturing processes, observations in the construction industry are still limited (see Section 7.1). Through the use of BIM data concerning the equipment and its position, an augmentation

can be displayed near or over its related equipment [25,72], helping the operator to better understand their link.

The data displayed can either be static or collected in real-time from the system (e.g., technical documentation of an equipment, time series of data collected by smart sensors, aand instructions for maintenance procedures) [26,43,53,55,64,67,68,70,72,73,77,78]. Most of the time, the selection of the data is done manually by an expert [25,53,69,73]. Once the selection is done, the data can be filtered, either using specific interactions through the XR device or thanks to a role specification of the operator [47] (see Figure 5a).

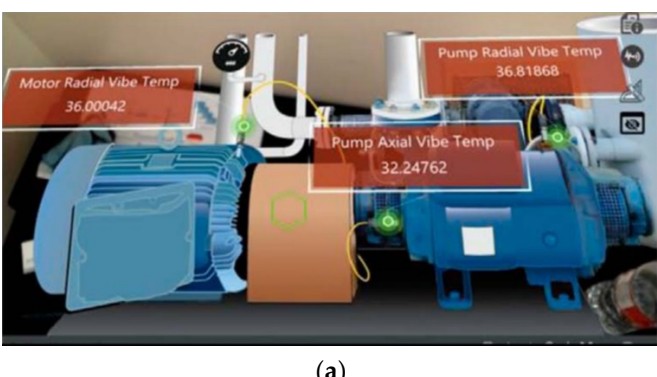

(**a**)

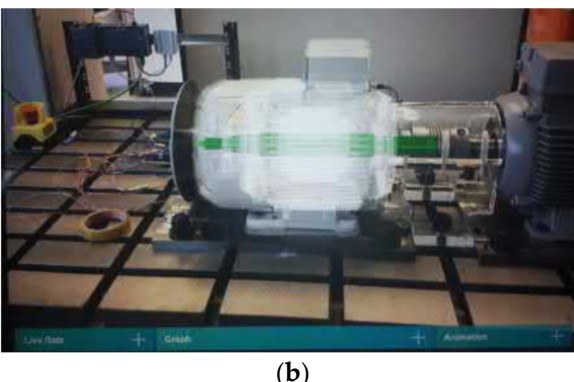

(**b**)

**Figure 5.** (**a**) Augmentations displayed to the operator during inspection of an asset [47]; (**b**) Healthy shaft (in green) of an electric motor [68].

A BIM-based DT contains the 3D models of the different equipment of a building. Some authors observed that they can be used to create immersive environments to promote collaboration between different distant users, such as for collaborative analysis of equipment through its digital representation [65,78]. These environments can be used to help with the creation of the augmentations [25,67,73]. One example is the *Corsican Twin*, where the user can create and attach the augmentations (e.g., specific documentation and time series) to 3D models of the equipment through the use of a virtual environment and immersive devices. It also allows specific areas to be defined near equipment so that when the operator using the AR or the MR device approaches the equipment, the augmentations will be displayed. This system is considered "*intuitive*" by its users because of the simulation of natural interaction that the immersive environment allows (e.g., drag-and-drop and click on an object). The data displayed here are time series as they are the most common in building management [25].

With the use of an AR or an MR device, the operator can see these 3D models overlaying their physical counterpart. This allows the operator to benefit from both the information needed for the procedure (e.g., real-time data feed and historical maintenance) and the contextual information from his or her surrounding environment. Once displayed, the 3D model can be enriched with the use of specific indicators to add information to the system. Such an indicator could be a change in the model colour to show if the system is behaving normally or unusually [25,43,53,68,76]. For example, Khalil et al. presented a solution to display the internal shaft of an electric motor [68]. When the shaft is red, it means that it is in a deteriorated state, otherwise the shaft is displayed in green (see Figure 5b).

### 5.2. Main Devices Used for Visualisation

There are two main types of XR supports that can be found in the literature (see Table 6). The first type is the handheld device (HHD) (see Figure 6a), such as a smartphone or a tablet [22,47,53,55,68,69,75,77,81], which is often already used in industry. Even if the HDD can be used to display augmentations, their main disadvantage is that they need to be supported, either by the user's hand or with a secure support. The second type of support is the head-mounted device (HMD). This type is mainly used to provide an immersive

environment for its user, leaving his or her hands free to interact with the real environment. There are two main subtypes of HMD that can be found in the literature: the optical and/or video see-through HMDs, which let the user see the real-word with immersive augmentations [25,26,43,47,66,67,70–73,75,78,80] (see Figure 6b), and the occluded HMDs, where the user is entirely immersed in a virtual environment [25,47,75,78,79] (see Figure 6c). While the occluded HMDs are mainly seen as VR devices, the see-through HMDs are mostly used for AR and MR applications. The most used optical see-through HMD in the literature is the Hololens [82], which provides the user both an immersive environment and intuitive interactions methods.

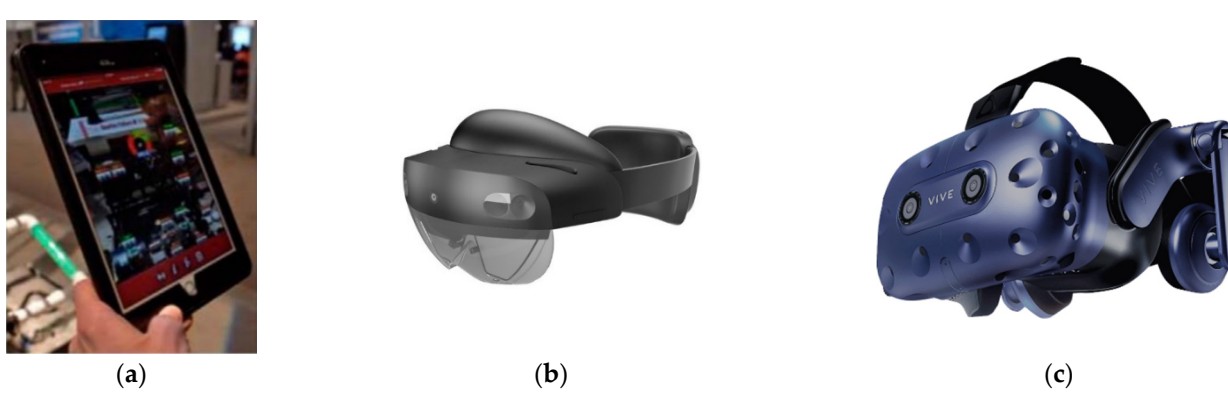

<div align="center">(<b>a</b>)      (<b>b</b>)      (<b>c</b>)</div>

**Figure 6.** (**a**) Handheld device [47]; (**b**) Occluded HMD example: HTC VIVE pro [83]; (**c**) See-through HMD example: Hololens 2 [82].

### 5.3. Interaction with the Model

A system as complex as a DT needs intuitive and user-friendly interactions for the user [25,26,77,84]. The most common interaction methods are through the use of standard inputs (e.g., touch screen and virtual keyboard for HHD, virtual buttons for HMD and smart-glasses) [25,70,76–78]. However, through the use of an embedded camera and specific sensors, the HMD can offer natural interaction methods to the user. Using image recognition algorithms and an embedded microphone, gesture and voice recognition can be performed by the user to provide natural interaction with the virtual interface [25,26,53,80]. However, in an industrial environment, external factors can disrupt these interactions (e.g., dust that disturbs image recognition and noise that complicates speech recognition). Other interaction methods can be used to overcome this problem, such as using the location of the user to display specific augmentations [25,47,75,76].

To locate the user in the real environment in comparison with the targeted equipment, some authors have decided to use the embedded GPS (Global Positioning System) of the devices (either HMD or HHD) [76]. The location it provides can then be used to filter the information to display, depending on the distance between him and the equipment or on the room he or she is in [25,47,75,76]. Prajapat et al. proposed a solution using a device's built-in GPS to display a simulated system configuration when the user is standing at a specific location [76]. However, the accuracy of GPS is too wide for effective use in indoor environments (precision of 5 to 10 m) [85]. Prouzeau et al. presented another example of the use of this interaction method by allowing the creation of specific zones in which the user must stand in order to view the information displayed to him [25] (see Figure 7).

Some authors have proposed using another interaction method to also estimate the orientation of the user, namely gaze tracking, which can be coupled with head tracking [25,66,71,72]. One of the main uses that can be made of gaze/head tracking is to select the augmentations that will be displayed, such as the time-series specific to a piece of equipment and which is displayed only when the user aims at it, as shown by Prouzeau et al. [25]. This method can also be used to interact directly with the displayed augmentations through the creation of a virtual point where the user's gaze impacts with the

augmentation (e.g., selecting the documentation to display with the Hololens by looking at a specific button [72]). The main advantage of this method is that it allows the user to interact with the digital environment, even if his or her hands are busy, or if the environment is too noisy for voice recognition. Some devices are emerging that directly integrate these features, such as Hololens 2 [82] and Vive Pro Eye [83].

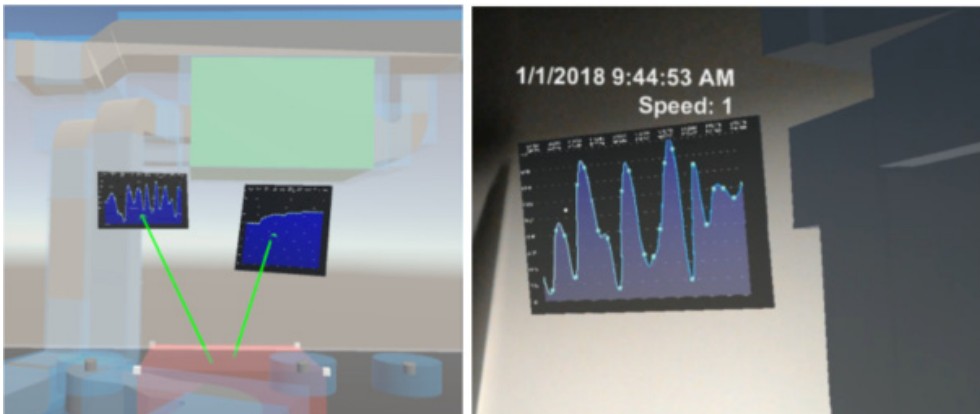

**Figure 7.** Example from Prouzeau et al.: On the left image, we can see the authoring of a specific zone (in red) for a situated visualisation. On the right image, the situated visualisation appears to the AR use when they enter the corresponding location [25].

Even if these interactions are mainly used to select the augmentations to be displayed (e.g., augmented instruction and real-time data of a specific equipment), they can also be used to take control of the physical equipment more precisely (e.g., controlling the trajectory of an industrial crane by looking at a digital representation of its end point [26].

### 5.4. Model Update and Collaboration

Through their different interaction methods, several complex processes can be reached intuitively. Some authors have decided to use the device's camera feed to collect various data using object recognition algorithms (e.g., segmentation algorithms applied to 3D point clouds [81], comparison with grid data from BIM [71]). Once an object has been identified, the information can be used for various purposes, such as to identify the discrepancies between the digital and real environment [71] (see Figure 8) or to link specific inspection feedback to the equipment's digital model [66].

Even though VR devices and common digital environments (CDE) are often used for evaluations of skills and training purposes, some authors have tried to show their benefits for collaboration [65,78]. CDE can be used to share information between different users with the digital representation of a piece of equipment, even if it is still in the conceptual phase. In this way, different experts can work together in a shared space, even if there are large distances between them. When the CDE is populated with BIM and equipment models, it can be used to facilitate the creation of the augmentations for AR and MR purposes. The VR user can create the augmentations in the virtual space and then define and observe how they will be displayed [25]. These CDEs could provide the designer with contextual information about the environment, such as the architecture of the building or the layout of equipment within the room. This information can be useful to avoid overloading the field of view while using AR/MR devices or to place augmentation near the correct equipment.

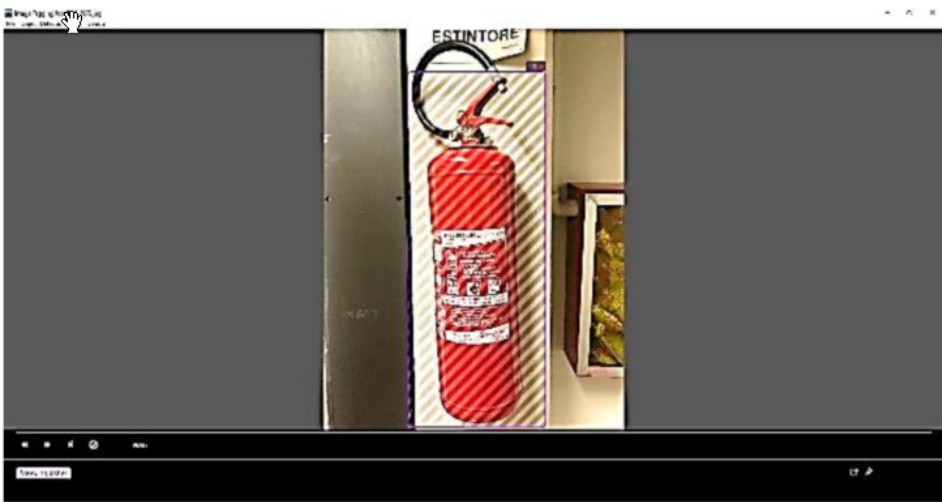

**Figure 8.** Image captured by the Hololens. A bounding box is drawn to delimit the object to label [71].

## 6. Maintenance Improvement Brought about by DT with XR Devices

While XR devices bring many improvements to the use of a BIM-based DT, some of them are more pertinent for maintenance operations. We identified 17 papers out of the initial total of 68 that focus on the combined use of XR devices and a DT to improve maintenance operations (Table 7). It can be observed that even if most of the related papers are use cases (56%), few of them in fact use a BIM-based DT (only 19%). In Table 7, the "*Visualisation*" part concerns all the information that can be displayed to assist maintenance operations but also to facilitate collaboration with a distant expert.

**Table 7.** Categorical review of maintenance-related papers using DT and XR technologies with/without BIM.

| Supported Activity | Features | Reference |
|---|---|---|
| Maintenance | Visual annotation<br>Track location<br>Ease inspection<br>Asset identification | [25,43,47,55,64,67–69,72,73]<br>[47,55,72]<br>[22,43,53,55,66,72,79]<br>[43,53,68,71] |
| Collaboration during maintenance procedures | Shared reports<br>Connected systems<br>Distant expert<br>Telepresence | [22,72]<br>[43,64,70]<br>[43,69,72]<br>[70,72,78] |
| Visualisation | Real-time, historical and documentation data<br>3D model<br>Filtered data | [22,25,43,47,53,55,68,70–72]<br>[43,47,67,69–72,79]<br>[43,47,68,72] |

### 6.1. Maintenance Improvements

When the operator is performing maintenance procedures on-site, it might be necessary to locate or identify specific equipment. XR devices have often been used to improve this *locating* task, using specific equipment data issued from a centralised DT or a BIM to detect its position [25,47,55,72]. Once the equipment location is known, XR devices can then be used to highlight it with, for example, a colour-coded 3D model [43,47,67,69–72]. For example, Xie et al. show the use of a Hololens headset combined with detection algorithms to facilitate the localisation and identification of malfunctioning elements [43]. Furthermore, the augmentations are not limited to the real-world obstacles. Even if the equipment is hidden or hard to see (e.g., behind a wall), a 3D model can be used to display it (see Figure 9). This simulated X-ray vision can also be used to highlight a specific part of a

piece of equipment, as shown by Khalil et al. in [68], which can be helpful to inspect specific subparts of complex equipment. It can also be used to display specific documentation, such as a 3D representation of an ultrasound inspection [72].

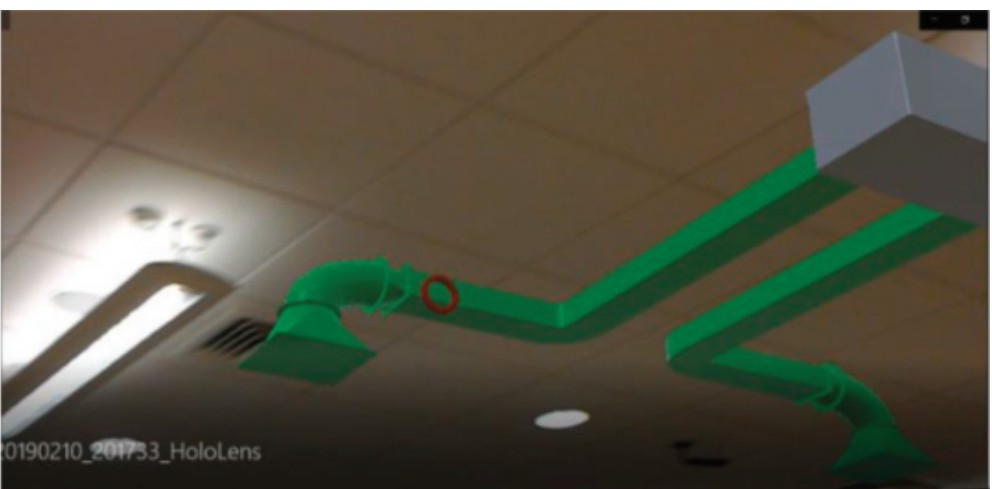

**Figure 9.** X-ray vision to display the 3D representation of an HVAC (heating, ventilation and air-conditioning) system hidden in the ceiling [43].

Some authors have observed that the combined use of centralised DT and XR devices can provide on-site operators with easy and intuitive access to the necessary data for maintenance procedures, such as equipment documentation or maintenance instructions [22,25,43,47,53,55,68,70–72]. Through the use of specific filters, both upstream and by the on-site operator, these data can also be specified to display some of them only for specific roles [47], but also through the use of the interaction methods offered by the XR devices (e.g., filter buttons, operator's location and operator's orientation) (see Section 5.3). Once the specification is done, the on-site operator will see only the needed information for the current procedure, preventing a mental load overflow due to too much information being displayed [86]. These augmentations can also help to display warning messages if needed to indicate either a failure or a security alert [64].

Once the information is chosen, various displaying methods have been proposed by the authors using XR devices. Most of them are based on the use of an AR or an MR device, thanks to its ability to overlay information onto the physical environment of the operator using augmentations (see Section 5.1). It has been observed in the literature that these augmentations can bring benefits to maintenance procedures. For example, during inspection of a fault [22,43,55,66,72], some authors have observed that the use of augmentations to display specific information over or near the equipment can help the operator to better identify the faulty equipment. This benefit also seemed to be reinforced when the operator's position and orientation, relative to that of the equipment, are considered to hide or display the information. Some authors have also observed that VR devices can be used to perform inspection in the virtual environment [79], allowing remote stakeholders to provide their expertise on it. The use of artificial lighting on a 3D representation of the inspected element could help to identify otherwise hard-to-see damage (see Figure 10).

Nevertheless, these benefits require that the positions of both the equipment and the operator are known. For the equipment, even if its position can be retrieved from a centralised DT, using a BIM or not, it is essential that this information is up to date, which is often not the case [25,28,33,43]. To prevent any issue, and to help with keeping the DT information up to date, authors have proposed the use of XR devices, and especially AR and MR devices, to analyse their users' surrounding environments and identify the visible equipment [43,68,71]. These projects can be used to help to identify specific equipment

to obtain linked information [71], to update the DT [81] or even to help with inspection feedback by linking data to the 3D model of the identified equipment [66].

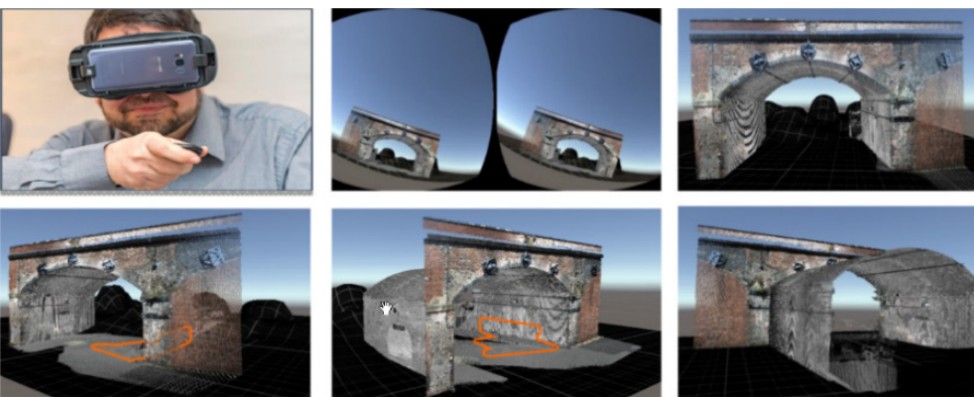

**Figure 10.** Inspection of a bridge 3D representation in a virtual environment [79].

*6.2. Synchronous and Asynchronous Collaboration during Maintenance Procedures*

Thanks to the centralisation of data made possible using a DT, stakeholders can participate more actively in the different phases of the project, whether they are on-site or on remote platforms. If the data are correctly filtered [36], every stakeholder is allowed to access and analyse their related data. Even before a maintenance operation is planned, the stakeholders are able to monitor the various equipment of a system and provide their own feedback on it. This information can then be added to its related equipment and eventually be used for predictive or inspection purposes (e.g., knowledge from expertise on a similar building used to estimate the remaining life) [22,43,70,72]. This *collaborative analysis* is considered *asynchronous* because it is carried out before the maintenance operations.

Occasionally, during maintenance procedures, the operator may require some assistance in resolving certain issues he or she may encounter. Some complex procedures may require contacting a remote expert for deeper information. Collaboration then becomes *synchronous* as it is carried out in parallel with maintenance operations. In the literature, there are multiple projects that promote this collaboration through the use of XR devices [22,43,69,72]. Using AR or MR devices, a remote expert can use a shared video feed to obtain contextual information concerning the performed procedure. Then, through the use of a digital avatar and a graphic representation of the gaze direction, the remote expert may simulate his or her presence to the on-site operator and thus share more accurate and contextual information [72] (see Figure 11). This collaboration promotes the sense of telepresence of the remote expert on-site [70,72,78]. The communication between the remote expert and the on-site operator can be performed with the combined use of a video call and augmented virtual annotations. These augmentations could be created in real-time by the expert, or before the maintenance, and triggered to provide information to the operator [25,43,47,55,67–69,72,73].

Once the maintenance is performed, the collaboration between the on-site operator and the remote expert could continue for deeper analysis. Thanks to the use of the centralised DT and augmentations through XR devices, inspection and maintenance reports could be shared between the on-site operator and multiple stakeholders [22,72]. However, research is still needed to propose a methodology to improve the way these reports can be created and sent to the DT by the on-site operator [66].

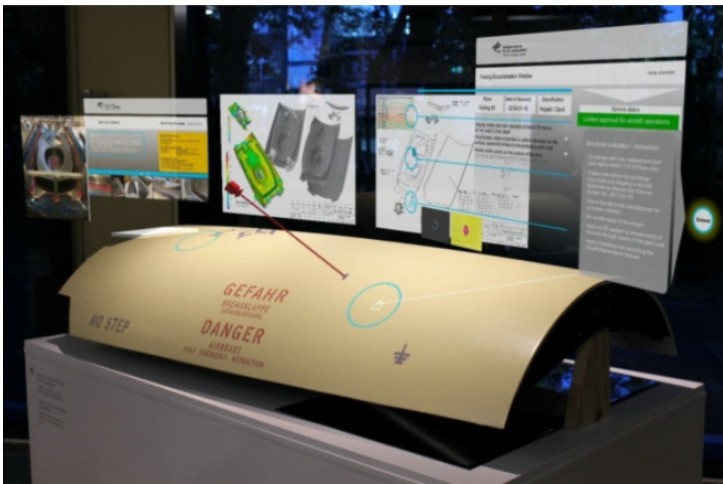

**Figure 11.** Collaborative inspection of an aircraft part. The expert's avatar is represented by the red box and his gaze by the red beam [72].

## 7. Discussion and Challenges

### 7.1. General Observations

Errandonea et al. have observed, in their review on the use of DT for maintenance, that the research in this domain is still recent [19]. In this literature review, we have made the same observation. Although our search dates back to 2005, we did not observe any studies before 2017 (see Figure 12), while the definition of Grieves dates from 2002 [17] and the one from Glaessgen and Stargel dates from 2012 [18]. However, we can observe a clear increase in the number of papers over the years since 2017. This increase can potentially be explained by the accelerating miniaturisation of computer systems, which makes it easier to set up smart monitoring systems [6,87].

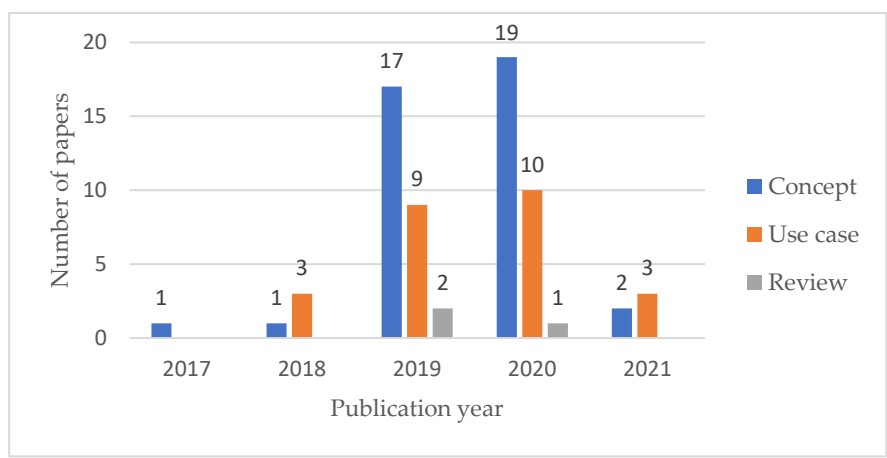

**Figure 12.** Type of paper by year.

To evaluate the advancement in research on the use of DT for building maintenance, we have decided to classify the resulting papers according to two main criteria. Firstly, we have separated the conceptual projects, use cases and reviews according to year (see Figure 12). We can observe that the number of use cases has increased over the years (from 3 in 2018 to 10 in 2020). However, the number of conceptual papers in the last five years shows how the research studies in this domain are still recent and how there is still a need to define clear implementation methodologies. The increase in use-case papers can also be explained by the improvements that have been done in the immersive technologies and on

their use to improve human–machine interactions [6,88]. Furthermore, *hyper-automation* has become increasingly common for industrial processes in recent years. This term defines the use of advanced technologies, such as artificial intelligence (AI) and machine learning (ML), to automate processes and provide advanced analytics [5]. *Hyper-automation* could then result in the creation of a DT of the entire organisation (e.g., roles, key performance indicators and processes).

Then, to observe how many *true* DTs have been developed in the last years, we have used the definitions of Kritzinger et al. to identify the evolution of the DT development by taking into consideration how the data are exchanged between the physical asset, its digital counterpart and the user. *True* DTs are those for which changes in the physical asset cause a change in its digital counterpart but also for which the physical is directly impacted by changes in the digital environment [21] (see Figures 12 and 13). Papers classified as "*undefined*", which are not reviews, do not define the whole of the DT development but only a part of it. Others focus more on the implementation of I4.0 technologies (e.g., IoT, BIM, XR) to improve the DT than on the development of the DT itself. It can be observed that the number of papers using *true* DTs has increased over the years (see Table 8), which can be explained by the improvements in IoT due to the miniaturisation of computer systems and to the improvements in interoperability between the equipment, the infrastructure and information systems [6,87]. However, only a third of these papers are use cases, most of them focusing on the development of concepts or new implementation methodologies (see Figure 14). Furthermore, even if the main scope of this research was the usage of DT in the construction industry and facility management, most of the papers found focused on projects related to the general industry (e.g., manufacturing, energy industry and aerospace) (see Figures 14 and 15).

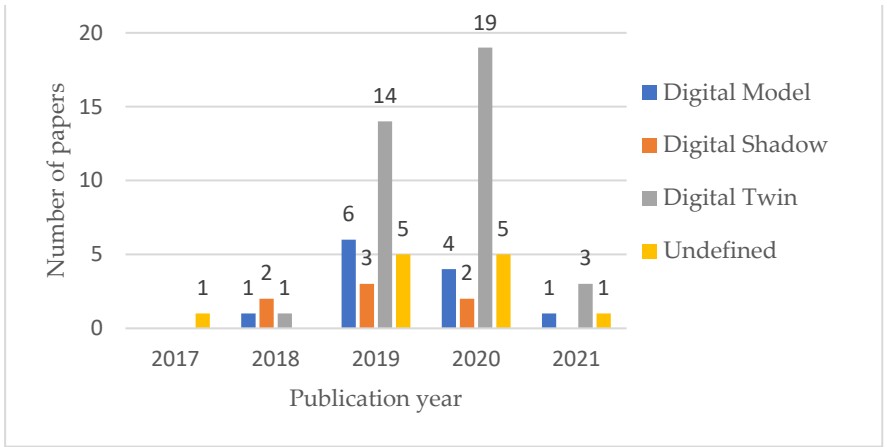

**Figure 13.** Number and type of DT by year.

**Table 8.** Number and type of DT by year.

| DT Type | Years | | | | | Total |
|---|---|---|---|---|---|---|
| | **2017** | **2018** | **2019** | **2020** | **2021** | |
| Digital Model | 0.00% | 1.47% | 8.82% | 5.88% | 1.47% | 17.65% |
| Digital Shadow | 0.00% | 2.94% | 4.41% | 2.94% | 0.00% | 10.29% |
| Digital Twin | 0.00% | 1.47% | 20.59% | 27.94% | 4.41% | 54.41% |
| Undefined | 1.47% | 0.00% | 7.35% | 7.35% | 1.47% | 17.65% |
| Total | 1.47% | 5.88% | 41.18% | 44.12% | 7.35% | 100.00% |

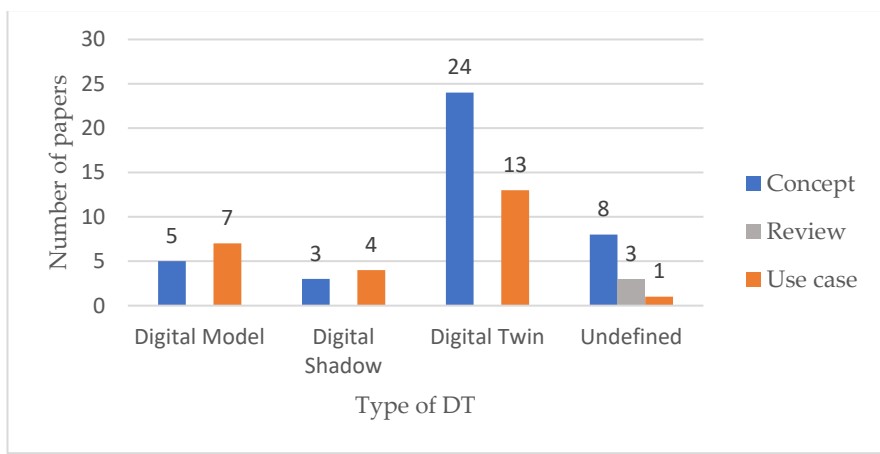

**Figure 14.** Type of paper for each type of DT.

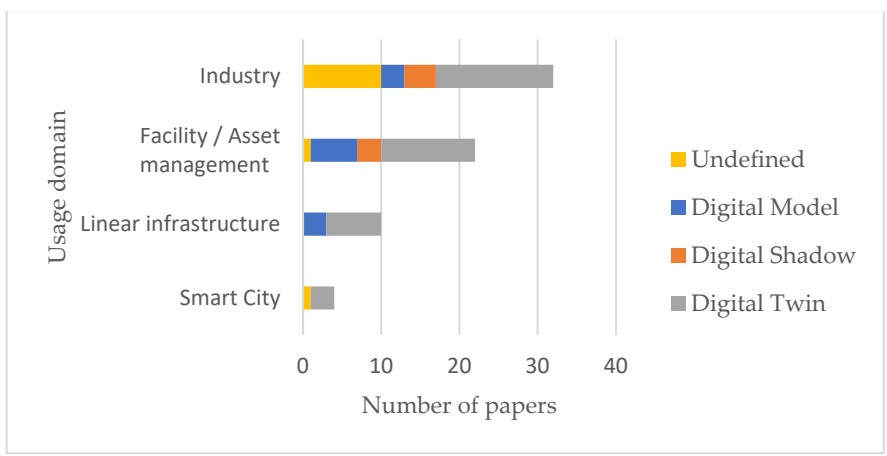

**Figure 15.** Main usage domains of the different DT types.

This lack of use cases and usage in building management shows that there is still a need to show the added value of using DT to professionals in the construction sector. It could explain why we have found only a few papers related to the application of a BIM-based DT combined with XR technologies for building maintenance.

We have also observed that some of the papers we found came from a project at the University of Cambridge called *Centre for Digital Built Britain (CDBB),* which aims to create a DT of the West Cambridge Research Facility [33,34,43,48,49,52]. Its work is part of a national United Kingdom project that aims to create a global digital framework for infrastructure data and to show the benefits of sharing quality information for the exploitation of these infrastructures. This project highlights the value of using a DT in building exploitation.

### 7.2. Future Usage: DT Interaction and Enhanced Maintenance Operations

In their proposed interaction framework based on XR technologies in DT, Ke et al. [75] have proposed a scheme to show the difference in interaction performance between VR, AR and MR. Their work has inspired us to create our own interaction scheme between the user, the DT and the application framework during maintenance operations (see Figure 16). The scheme is composed as follows:

- The Digital Twin is represented by three parts:
  - ○ *Physical part*: This represents the physical asset of the DT (e.g., equipment, building or equipment). Data can be sent from and to this part thanks to the *data processing* part. However, if on-site, the *User* can also interact directly with it, especially during maintenance operations or visual inspections.
  - ○ *Digital part*: This is the digital representation of the asset, with the semantic and real-time data gathered with smart sensors and processed by the *data processing* part. The data can then be linked to the 3D representation of the asset and then displayed to the user once processed in the appropriate language.
  - ○ *Data processing*: This contains all the communication protocols allowing exchanges between physical and digital parts, and the algorithms that process the raw data before sending it to the digital part. It also contains the decision and prediction algorithms as well as the ones allowing the transmission of the user's commands to the digital part to display information and to the physical part for the control of the real system.
- *HMI*: The human–machine interface (HMI) part represents the communication interface between the user/expert and the DT (e.g., Information displayed and equipment's controls). This interface can either represent an XR device or a classic computer software allowing 3D information to be visualised thanks to the *data processing* part that translates the commands into the appropriate language. It can also represent VR training applications using BIM-based DT to train the user on realistic situations through an immersive environment and collected data from real-life situations [78,89].
- *User/Expert*: The on-site user can interact with the DT thanks to both the *HMI* part and direct interaction with the *Physical* part, especially during maintenance operations. On the other hand, the remote expert can interact with the DT only by using an *HMI*, as the distance prevents any direct interaction with the *Physical* part. The blue two-way arrow represents the collaboration that can happes between the on-site user and the remote expert when needed, either using the *HMI* part (such as a specific MR application) or with an external device (such as a phone call).

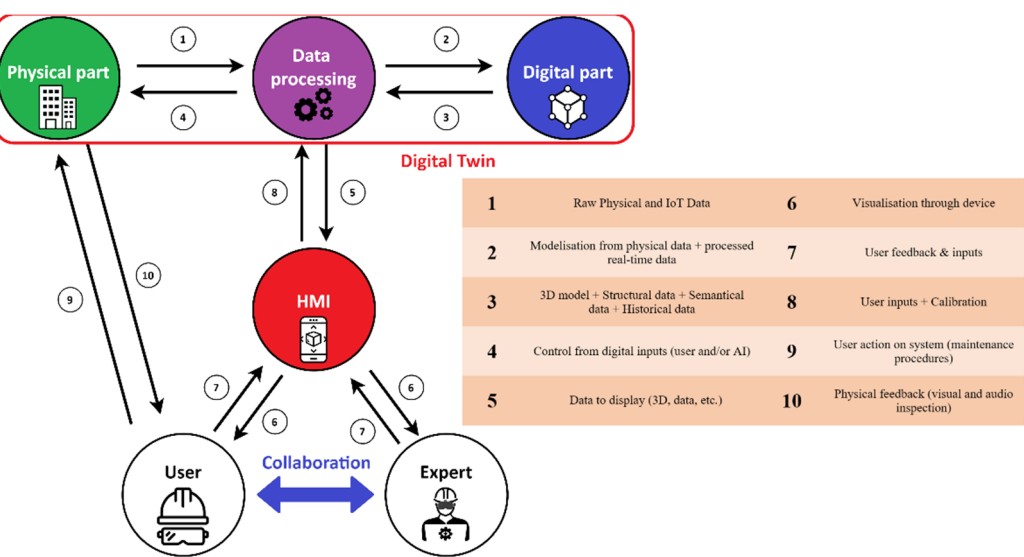

**Figure 16.** Interactions for an on-site user and a remote expert.

### 7.3. Challenges

The lack of use cases has already been observed by some authors [28,29,33,34,51,78]. Various factors can explain this difficulty. In addition to technical challenges, such as data management and issues related to the use of XR devices, multiple authors have observed

that changes are needed in the organisation and management of maintenance procedures for a correct implementation (see Table 9).

**Table 9.** Categorical review of the new needs for correct implementation.

| Category | Challenges | Reference |
|---|---|---|
| Organisational, human and economic changes | Implementation | [29,41,90] |
| | Data selection | [25,29,33,35,36,47,91] |
| | Information on financial risks | [35,90] |
| | New technical skills | [78,86,91] |
| | Define roles | [36,47,92] |
| | Collaboration management | [16,22,25,29,32,33,38,40,41,65, 69,70,72,78,79] |
| Data management | Lack of data for DT creation and update | [21,32,37,38,49,61,74] |
| | Lack of data standards | [28,33–35] |
| | Data ownership | [26,31,37,51,54,92] |
| | Data security | [31,35,46,54,64,72] |
| Challenges in using XR devices | Ergonomic and physiological issues | [25,84] |
| | Calibration | [25,26,53,66,71,72,77,80,81] |

### 7.3.1. Organisational, Human and Economic Changes

The O&M phase is one of the most expensive of the building lifecycle [19]; however, there is a lack of evidence in the literature of any real reduction in these costs [40,90]. Furthermore, authors have observed that most of the professionals are not fully aware of the initial costs and risks involved in implementing a DT, nor are they aware of the amount of initial work needed [29,47,90]. Therefore, there is a need to provide clearer and more precise evidence of the financial benefits that these technologies can bring in the long term and thus to establish use-cases that take into account the initial implementation costs. Love and Matthews have also observed that, while most of the papers are focused on *why* the professionals should implement these technologies, few explain *how* these can be implemented in their existing processes [90]. Some changes are needed in the organisational and managerial structure of the O&M phase when it is decided to implement these technologies [29,41,47,90], such as the early involvement of those responsible for the operation to identify the information that will need to be retained throughout the life of the building.

Many authors have observed the need for early consideration of maintenance needs [25,29,33,35,36,47,91]. It seems that the O&M phase is often overlooked during the building design and construction. However, these phases contain many useful data for the maintenance process (e.g., geometrical information, equipment placement, construction modifications) [28–30,35,40,42]. BIM can be used in those phases to retrieve the information needed in an intuitive way, such as linking the information to the 3D representation of the equipment to access it by interacting with the 3D model [28–30]. However, even when the BIM is used to create a DT, it is still needed to allow communication between the actors of the different phases to define which information must be transferred throughout the building lifecycle [35]. However, this can be complicated because the actors involved in the operation of a building may not yet have been defined when the design or construction phase was started. Thus, it may be interesting to consider a method to automatically define the essential data for the whole life of the building. This definition may be done using AI or ML algorithms fed with known maintenance information from previous, existing buildings.

It has been observed that the adoption of new technologies in existing processes can be complicated for users [78]. It is therefore necessary to create new strategies to implement them [41,86]. Several of the concept papers we retrieved in this research proposed methodologies for implementing DT and the various I4.0 technologies

[22,28,29,33,34,36,39,41,42,45,46,51,57,62,64,81,86,90,93]. For example, Moretti et al. proposed a methodology to implement a BIM-based DT on a building with incomplete as-built data [34]. Another example is given by Bevilacqua et al. where they present the different criticalities that need to be taken into account when developing and implementing DT and XR devices to prevent operators' risk during the O&M phase [64]. However, some authors have also observed that the introduction of these new technologies creates a need for new skills to allow the operators to use them. This Operator 4.0 [91] does not need to create and capture the data, thanks to the introduction of smart and embedded sensors. Therefore, they need to develop new analytics skills that will help them to perform advanced operations using the collected data [36,78,91]. However, the operators do not need to have all the analytics skills. Some authors have therefore observed that there is a need to define the specific role of an operator and use this classification to display only specific information in the function of the user's role [36,47], further improving the security and the privacy of the shared data. This change in the role of the collaborators reminds us of the importance of human involvement in operations. This importance was highlighted in 2020 to define the new industrial revolution: Industry 5.0 [94].

The need to change the maintenance organisation is even more important with the impact of the COVID-19. It is now necessary to think about new methods of remote collaboration, whether through the centralised source that is the DT but also through the use of XR technologies [16,22,25,29,32,33,38,40,41,65,69,70,72,78,79]. This collaboration can be achieved using an MR device, as proposed by Utzig et al. for aircraft maintenance tasks [72]. It can also be performed in a collaborative virtual environment accessible with a VR device where users can work together on the 3D representation of a piece of equipment, as shown by Omer et al. for inspecting a 3D representation of a bridge [79]. Novack et al. also observed that this collaboration in virtual space can also allow for collaborative work on the design of a system, such as a building, even if the stakeholders cannot move in reality, by allowing them to observe it at different scales [78].

### 7.3.2. Data Management

Other challenges that have been observed in the literature concern the definition of DT and data management. As observed by Kritzinger et al., the definition of DT varies between different authors [21]. Kaewunruen et al., to name but a few, consider BIM as being a DT, even without the automatic exchange of data [32,37,38]. It has also been observed that the problem of DT maintenance is often overlooked. Pileggi et al. proposed a methodology to perform a continuous update of the DT [74], but their methodology assumed that all the needed data exist. Some authors have already observed that, even at its creation, there is a clear lack of necessary data to create the DT [49,61]. Therefore, the data needed to update it can be missing or could be impossible to retrieve.

One of the first causes of difficulty in the retrieval of data is the different proprietary formats that exist. Each IoT system may have its own proprietary format, which can complicate exchanges between them [35]. Multiple authors have observed that, for better interoperability, a common format needs to be defined for data exchange between the different stakeholders [28,33,35]. The IFC format has already been defined as one of the main formats to structure a BIM, but it is still needed to adapt the data coming from smart systems and smart sensors. Some authors, such as Moretti et al., have proposed an open-source methodology based on IFC format to improve the IoT integration, even with incomplete data, into a BIM [34]. Their method used predefined common metadata attributes to add data from heterogeneous sources homogeneously through various IFC objects. Such data standardisation may also be necessary to perform inspection feedback in an efficient way. Using DT and XR technologies, the on-site operator may provide advanced inspection feedback, through both textual and vocal information, but also with real-time data provided by IoT devices. For example, Kunnen et al. proposed a methodology to add inspection data directly to the 3D model of equipment using gaze-tracking and image recognition, which can facilitate future retrieval of information [66]. However, there is

still a need to define a methodology to help feed back professional knowledge specific to maintenance operations within the DT, such as maintenance procedures or personal observations [66,95].

Another challenge that has been observed in the literature is that there is no clear approach concerning data and DT ownership. Multiple authors have asked and studied this question in recent years [26,31,37,51,54,92]. This question is particularly important as it defines who is supposed to oversee the implementation, the maintenance and the update of the DT. Throughout the building lifecycle, several different companies may be involved, from its design, through its exploitation, to its disposal or renovation. Thus, there is a question of who owns the data collected during the building's lifecycle. Furthermore, as a BIM-based DT is supposed to allow access to all of the building information, the question of its owner can also be asked [26]. It is also necessary to define which data can be shared and which cannot, as some companies can collect sensitive data [31,47,54]. This point brings up the issue concerning the data security, especially during the exchange of data between the systems and its visualisation by the users [31,35,46,54,64,72]. It may be interesting to develop an encryption protocol for communications between a DT's different components, for example. Some authors have also observed that the use of blockchain technology, initially created for Bitcoin, is developing in the context of Industry 4.0 and could address the issue of privacy and security of data exchange with a DT. The ownership of the data and the communication protocols must therefore be defined early enough to allow a clear and smooth development of the DT but also to allow the overall cost of its creation to be defined [5,6,53,96].

### 7.3.3. Challenges in Using XR Devices

In recent years, with the improvements in and miniaturisation of the computer, the development of XR technologies has increased, and new challenges have arisen with it [5,6,75,97]. A lot of research is underway to improve these technologies and speed up their implementation.

Firstly, several issues concerning the usage of HMD have been found in the literature, mostly through feedbacks from users [25,84]. The comfort of the device is a major concern for most users, especially during continuous use (e.g., the field of view's size and weight). *Cybersickness* is also an important aspect that needs to be taken into account when using XR devices [84]. It is caused when visual, vestibular and proprioceptive systems conflict with each other, in which the eyes perceive a movement that is not synchronised with what the motionless user perceives. The main symptoms are headaches, disorientation and nausea. It can also be caused by ergonomic discomforts, which reinforces the importance of the right choice of the XR device.

To correctly apply the augmentations in the virtual space, a good calibration and pose estimation is needed. Many authors have tried to resolve this issue by using the object recognition abilities offered by the camera of XR devices [66,71,81], but most of the literature uses specific markers to define what to display or which equipment the user is targeting [25,26,53,71,72,77,80]. Nevertheless, regardless of the initial calibration method used to display the augmentations, the precision of this calibration needs to be maintained throughout the use of the solution [72]. Prouzeau et al. propose an alternative by allowing the designer of the augmentations to fix their position near their related equipment directly in the virtual space [25]. Once the user has performed the initial calibration to detect in which room of a building he or she is, the augmentations is enabled through specific triggers and their position is fixed even if the users move in the calibrated room.

## 8. Conclusions and Perspectives

### 8.1. Conclusions

In this review, we have observed that most of the advantages of using DT and XR technologies to improve industrial performance can also be applied to improve building exploitation, especially when using data extracted from the BIM of the building to create

the DT. A BIM-based DT can be likened to a centralised database where the real-time collected data of an equipment and its static data, such as maintenance history and technical documentation, are linked to its 3D representation, which facilitates its retrieval when needed. During the exploitation of the building, these data can then be used to monitor the building and its equipment, improve inspection operations while facilitating access to equipment data, and use the prediction algorithms of the DT to plan the maintenance when needed.

XR technologies can be used to visualise the equipment data directly in front of the equipment, but also specific information either provided by an expert or automatically added by an algorithm. These devices also allow for better data management using enhanced interaction methods such as gesture and voice recognition. Their built-in camera can also be used for image recognition to identify the inspected equipment. This information can then be used to detect discrepancies between the physical and the digital environments, but also to link inspection feedback to its digital representation [66].

With the impact of COVID-19, it is now necessary to provide new methods to facilitate collaboration with a remote expert when needed [97]. Either using virtual annotations or a shared video, the XR devices provide remote experts with contextual information and advanced methods to display information to the on-site user. The position and the orientation of the on-site user in relation to the inspected equipment can also be shared with a remote expert via a digital avatar. Another aspect of this collaboration is the use of the centralised BIM-based DT, where various remote experts can provide their own data analysis and add meaningful information for maintenance procedures.

Furthermore, DT was originally been proposed to show the real-time evolution of an asset. Mainly used in the manufacturing industry, this review shows that it can be expanded to represent the twin of an entire smart building. However, another concept has been emerging in recent last years: the concept of a *Smart City*. Some authors have already investigated how a city-scale DT can be created [30,31,52,62] and used for individualisation [22] to specify the city assets, such as buildings.

*8.2. Perspectives*

In this review, we have observed that a BIM-based DT can be useful for maintenance operations, from planning to implementation. Thanks to the use of XR devices, the maintenance inspection process can be further improved with the display of data linked to the inspected equipment. Furthermore, a BIM-based DT can be used to provide access to the entire system architecture and thus provide meaningful data related to the upstream and downstream equipment. Such information could be useful to identify the existing cause-and-effect relationship and thus prevent a possible ripple effect in the case of failure while also identifying potential origin of one.

However, these technologies can also be used to provide more meaningful feedback on the inspection. The camera built into XR devices, combined with their gesture and voice recognition, could be used by operators to provide contextual information. With easy access to BIM-based DT, operators could also add information from equipment data retrieved over a specific period. This information can later be used by another operator to improve future inspections or even by the DT's decision algorithms to allow early identification of future failures.

**Author Contributions:** Conceptualization, C.C., S.N., P.R., D.B. (David Baudry), D.B. (David Bigaud); Data curation, C.C.; Literature search and review, C.C.; Methodology, C.C., S.N., P.R., D.B. (David Baudry), D.B. (David Bigaud); Supervision, S.N., P.R., D.B. (David Baudry), D.B. (David Bigaud); Validation, C.C., S.N., P.R., D.B. (David Baudry), D.B. (David Bigaud); Visualization, C.C., S.N., D.B. (David Baudry); Writing—Original Draft, C.C.; Writing—Review & Editing, S.N., P.R., D.B. (David Baudry), D.B. (David Bigaud). All authors have read and agreed to the published version of the manuscript.

**Funding:** This work is supported by a doctoral grant from the University of Angers and the CESI Group in France.

**Informed Consent Statement:** Not applicable.

**Acknowledgments:** The authors would like to thank Polytech Angers, Angers, France and CESI, Le Mans, France for their support throughout the research project.

**Conflicts of Interest:** The authors declare no conflict of interest.

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
