# Peer review of "BIM-Based Digital Twin and XR Devices to Improve Maintenance Procedures in Smart Buildings: A Literature Review"

_applsci, doi:10.3390/app11156810_

Round 1
Reviewer 1 Report
The literature review carried out brings the reader closer to how a Building Information Modeling can be used to create a Digital Twin and what opportunities this offers when combined with extended reality technologies
The authors draw attention to the important issue of data collection not only for designed buildings but also for existing buildings and the use of this data, to maintenance and operate facilities.
Based on the authors' analysis, it can be noted that despite the development of algorithms, there is still little in the scientific literature about the use of the methods in practice. The developed review indicates the potential of scientific research on the use of the discussed technologies in practice.

Reviewer 2 Report
This review paper presents a comprehensive survey paper on studies using BIM and XR devices.
I have a few comments.
- 'Digital Twin' often refers to making a digital copy of the real world. Then changes in the real world are reflected on the digital copy (the virtualized model), and the changes in the virtual model are also reflected on the real world (in forms of sensor/device operation, movements in physical actuators). I think this survey paper lacks in the second part (the real-world changes to be reflected in the virtual model).
- A very similar approach/topic is covered in a recently published paper. "Singh M, Fuenmayor E, Hinchy EP, Qiao Y, Murray N, Devine D. Digital Twin: Origin to Future. Applied System Innovation. 2021; 4(2):36. https://doi.org/10.3390/asi4020036". I suggest the authors take a look at this paper and highlight differences in the proposed work.
- Several "Error! Reference source not found." Please correct those errors.
- Previous studies on DT have different scales of DT. Some are building-levels while some studies are only applicable to room-level or objects. I would like the authors' own definition of DT should include characteristics of "scalability". Also, this scalability may be used to exclude or include more proper on DT (for example, exclude DT concepts smaller than room-level). Lastly, discussion and challenges regarding "scalability" should be addressed.
Reviewer 3 Report
See the attached file.
